# Topical Application of No-Ozone Cold Plasma in Combination with Vitamin C Reduced Skin Redness and Pigmentation of UV-Irradiated Mice

**DOI:** 10.3390/biomedicines11061563

**Published:** 2023-05-27

**Authors:** Seoul-Hee Nam, Jeong-Hae Choi, Gyoo-Cheon Kim

**Affiliations:** 1Department of Dental Hygiene, Kangwon National University, Samcheok 25913, Republic of Korea; nshee@kangwon.ac.kr; 2Research and Development Center, Feagle Co., Ltd., Yangsan 50614, Republic of Korea; 3Department of Oral Anatomy, School of Dentistry, Pusan National University, Yangsan 50612, Republic of Korea

**Keywords:** no-ozone cold plasma, ultraviolet B, skin whitening, Vitamin C, tyrosinase inhibitory

## Abstract

Ultraviolet (UV) is the main cause of sunburn on the skin as it induces erythema and accelerates pigmentation. Vitamin C is one of the most frequently used compounds to reduce UV-induced skin pigmentation, but it has limitations in absorption through the skin. In this study, we tested whether a no-ozone cold plasma (NCP) treatment can improve UV-irradiated skin by helping the action of Vitamin C. For this, among five groups of HRM-2 hairless mice, four groups of mice were subjected to UVB irradiation, and three groups of UVB-treated mice were treated with NCP, Vitamin C, and NCP + Vitamin C, respectively. For evaluating the effect of each treatment, the melanin and erythema index was measured during animal experiments. Histological changes were monitored by performing H&E and MTS and IHC against tyrosinase and melanin. As a result, the naturally recovered mice showed a 28-point decrease in the melanin index, whereas a decrease of around 88, 74.3, and 106 points was detected in NCP-, Vitamin C-, and NCP + vitamin C-treated mice, respectively. Likewise, only a 39-point reduction in the erythema index was monitored in naturally recovered mice, but the NCP-, vitamin C-, and NCP + vitamin C-treated mice showed a 87.3-, 77-, and 111-point reduction, respectively. Interestingly, the skin tissues of the mice treated with NCP in combination with Vitamin C mostly recovered from UVB-induced damage. Altogether, this study elucidated the beneficial effect of the treatment of NCP in combination with Vitamin C on the UVB-irradiated skin, which might be helpful for treating sunburn on the skin.

## 1. Introduction

Human skin is constantly exposed to ultraviolet (UV) radiation in sunlight [1]. Although UV radiation has beneficial effects, such as vitamin D synthesis [2], UV radiation can not only modulate the function and survival of several type of cells but also acts as a key factor for inducing skin cancer [3]. Melanin is not only one of key factors for protecting the skin from UV rays but is also an important factor that determines skin color [4]. Melanin synthesis in the human epidermis occurs within melanosomes of the melanocyte [5]. When the skin receives sunlight, the UV-damaged keratinocytes transfer their stress to nearby melanocytes, and this leads to melanin synthesis. After melanin synthesis, the melanin is packaged into melanosomes, which are exported to keratinocytes, where they localize over the nucleus. Through this melanin synthesis procedure, skin can be protected from further UVR-induced damage but skin color can also be darkened. Sunspots and freckles are also pigmented through excessive melanin generated by UV rays [6]. Skin abnormalities caused by the abnormal pigmentation of sunspots and freckles lesion can also lead to social phenomena including loss of self-esteem, social phobias, and depression [7].

Today, as the demand for white and transparent skin increases in cosmetic market of Asia, the demand for skin whitening products increases. Skin whitening cosmetics should have the function of suppressing melanin pigment production or decolorizing by reducing generated melanin [8]. Vitamin C (Vt. C), also called ascorbic acid, is the most powerful and broad-spectrum antioxidant among the antioxidant vitamins in cells. Vt. C not only removes free radicals by preventing oxygen from reacting with them but also prevents oxidation-induced DNA mutations in human cells [9]. On the skin, Vt. C has effects such as anti-aging, whitening, moisture retention, giving vitality to tired skin, protecting the skin, and promoting collagen and elastin synthesis. Furthermore, Vt. C is used as a treatment method in the depigmentation of hyperpigmented spots on the skin [10]. Given the acidic and antioxidant properties of Vt. C, the use of Vt. C can block the synthesis of melanin [11]. However, since Vt. C is water-soluble, it has limitations; for instance, the topical application of Vt. C hardly penetrates the skin [12]. Although it is well known that the controlled laser ablation of stratum corneum can improve Vt. C penetration into the skin, this treatment needs to be performed under strictly controlled conditions [13]. Furthermore, since the demand for non-ablating, non-painful techniques for delivering effective ingredients into the skin is still increasing, a new technique for delivering Vt. C is required.

Plasma is the fourth state of matter after solid, liquid, and gas, and is defined as an ionized gas [14]. Plasma is widely applied in the medical field, and the medical properties of plasma, such as safe sterilization, anti-cancer, skin regeneration, and teeth whitening are highly anticipated in terms of utilization and effectiveness [15,16,17,18]. In our previous studies, we introduced no-ozone cold plasma (NCP) technology, which was developed for the safe use of plasma in the human body. To date, the beneficial effects of NCP on the skin, such as the enhancement of transdermal drug delivery [14], skin renewal activity [19], and anti-inflammatory activity in an atopic dermatitis mice model [20], have been reported. However, there are no studies on the effects of NCP on skin pigmentation caused by UV irradiation. To prove the beneficial effect of NCP on UV-pigmented skin, the skin-recovering effect of NCP on UVB-irradiated skin was tested, and its effectiveness was compared with the effects of vitamin C. Furthermore, the combinational treatment of NCP with vitamin C was also carried out in order to prove that NCP treatment can be helpful for maximizing the effect of vitamin C and quickening the recovery of UVB-irradiated skin.

## 2. Materials and Methods

### 2.1. NCP Device for the Animal Experiments

The NCP device was developed by the FEAGLE corporation (Yangsan, Republic of Korea) for dermatological research and introduced in our previous studies [19,20]. In brief, the device has a main body, which is composed of an SMPS (switching mode power supply), solenoid valve, gas flow rate controller, high voltage circuit, among other things, and it also has a handpiece, which is the part that generates the NCP. Inside the handpiece (Figure 1B), there is a coaxial DBD (dielectric barrier discharge) type plasma source, which consists of a stainless steel (stainless steel 306) inner electrode and an outer electrode (Al_2_O_3_) surrounding the outer diameter of the ceramic nozzle. When the start button on the main body is pressed, argon gas flows between the inner electrode and the ceramic nozzle, and an output voltage of 3 kVpp (voltage peak to peak) with a frequency of 20 kHz is applied to the electrodes to generate plasma. The argon gas rate of this device can be regulated through gas flow rate controller, and in this study, the gas flow rate was fixed at 1 slm (standard liter per min). Detailed information on the physiochemical properties of the NCP can be found in our previous study [17]. In brief, NCP does not create ozone and nitrogen oxides due to the unique structure of the plasma-generating part of the device, and only argon ions and electrons and a few OH radicals are generated. Since the plasma plume of NCP forms in the plasma generating module only, the afterglow plasma gas of about 27 °C can be treated on mice skin. The distance of the surface of mice skin from the electrodes was kept as 1 cm during the treatment.

### 2.2. Animal Model and Skin-Whitening Procedure

Male HRM-2 melanin-possessing hairless mice aged 5 weeks were obtained from the Central Laboratory Animal Inc. (Seoul, Korea), and housed in a controlled 12 h light/dark cycle at a temperature of 22 ± 1 °C and relative humidity of 50 ± 5%. The animal experimentation was performed in accordance with a protocol approved by the Institutional Animal Care and Use Committee, PNU (NTU PNU-2022-0225). Reflecting the accepted rate for minimizing the number of sacrificed animals, the proportion of animals dropping out due to shock, etc., during the experiment was calculated as 20%. Accordingly, a total of 30 mice were divided into 5 groups (*n* = 6).

As described in Figure 1A, all groups of mice (6 weeks old, about 25 g of weight) were administered with general anesthesia comprising 0.5 mL Avertin (1.25% 2,2,2-tribromomethanol, 2.5% 2-methyl-2-butanol dissolved in ddH_2_O), which was intraperitoneally injected into a 250 mg/kg mouse as an anesthetic agent. The mice in 4 experimental groups (groups 2~5) except for the control group (group 1) were irradiated with 300 mJ/cm^2^ of UVB 3 times in a bi-daily manner. Immediately after the last UVB irradiation (Wednesday, day 1), the dorsal skin above the tail region (the region indicated as the round circle in Figure 2) was subjected to the measurement of melanin and the erythema index (MI and EI), and then treated with each treatment method. For group 2, the target dorsal skin was subjected to treatment with a 1 × 1 cm^2^ sized patch containing phosphate-buffered saline (PBS) for 5 min. For group 3, the mice skin was treated with a patch containing 10% vitamin C (Vt. C; L-Ascorbic acid 99%, Sigma-Aldrich, St. Louis, MO, USA) for 5 min. For group 4, the skin was treated with NCP for 5 min, and a patch with PBS was placed on the same site for 5 min. For group 5, the skin was treated with NCP for 5 min, and then a 10% Vt. C patch was placed for 5 min. All treatment procedures were performed on Wednesday (day 1), Friday (day 3), and Monday (day 6), 3 times in total. On day 7 (Tuesday), all mice were sacrificed in a CO_2_ gas chamber, and the treated dorsal skin tissues were isolated and used for histological evaluation.

### 2.3. Measurement of Skin Conditions

The changes in the appearance of the skin during animal experiments were observed using a digital camera. Photographs were taken using a digital camera to document the skin lightening under standardized lighting conditions, distance, and exposure, namely with blackout curtains, a fixed distance, and an exposure time of 90.0 ms, with constant temperature conditions (24–26 °C) and air humidity (33–41%). The MI and EI of the target dorsal skin (the circle region in Figure 2) were measured using a Mexameter^®^ MX18 (Courage + Khazaka electronic GmbH, Köln, Germany) and a contact probe connected to the body in the Multi Probe Adapter^®^ MPA 5 Systems (Courage + Khazaka electronic GmbH, Köln, Germany). The measurement was performed at 4 time points, just before the 1st treatments (before), immediately after the first treatment on day 1 (1st), after the second treatment on day 3, and immediately after the third treatment on day 6. To calculate ΔE MI and EI value, the MI and EI value of each mouse on day 1, day 3, and day 6 were subtracted from the MI and EI value measured at just before the 1st treatment (before).

### 2.4. Histological and Immunohistochemical Analysis

The skin samples from each group were fixed overnight with 4% paraformaldehyde at room temperature for 24 h. The fixed tissues were embedded in paraffin under vacuum, and sections measuring 4 μm were deparaffinized and rehydrated. Hematoxylin and eosin (H&E) staining was performed to evaluate epidermal or dermal changes and morphological changes in the tissue. Masson’s trichrome staining (MTS), or three-color staining in histology; cell nuclei (dark red), collagen (blue), and cell cytoplasm (red/purple), was used to examine the epithelial and epidermal structures in the mouse’s skin. The collagen fibers, nuclei, and background were stained blue, black, and red, respectively. For immunohistochemical (IHC) analysis, the tissue sections were incubated with antibodies against the tyrosinase antibody (diluted 1:200; sc-15341, Santa Cruz, CA, USA) overnight at 4 °C. After washing, sections were incubated for 30 min with peroxidase-labeled polymer-horseradish peroxidase (HRP), conjugated to goat anti-rabbit immunoglobulins (Envision + System-HRP-DAB; Dako, Carpinteria, CA, USA) for 1 h. Staining was completed via incubation with 3,3′diaminobenzidine chromogen solution (chromogen solution is part of the Envision kit). The sections were counterstained with Harris’ hematoxylin, dehydrated, and mounted on a cover slip. In addition to histochemical stains, Fontana–Masson staining was used to visualize the melanin pigment using a Fontana–Masson Stain (FMS) Kit (ab150669, Abcam, Cambridge, UK), according to the manufacturer’s instructions. A reduction in ammoniacal silver nitrate to metallic silver shows melanin turning from brown to black and reveals nuclei as pink to red. The staining was analyzed using a BX51 microscope (Olympus; Tokyo, Japan), equipped with a ×40 objective. Images were taken using a digital camera (Pixel link PL-B686 CU, Ottawa, ON, Canada).

### 2.5. Statistical Analysis

Statistical analysis was carried out using SPSS Ver. 26 (SPSS Inc., Chicago, IL, USA) software. The melanin and erythema indices, the results of quantification of collagen, and the tyrosinase level from the histological images were analyzed using one-way analysis of variance (ANOVA) and Ducan’s post hoc test. Since the normality of the data was not secured, it was analyzed after conversion to logarithmic values. The significance level of the statistical test was considered 0.05.

## 3. Results

### 3.1. The Combined Treatment of NCP and Vitamin C Revealed the Fastest Restoring Activity in the Appearance of the UVB-Irradiated Skin

To visualize the effect of NCP in combination with the Vt. C on the external change of UVB-irradiated skin, the photographs of the dorsal skin at the end of the animal experiments were taken, and the representative photograph of each group is presented in Figure 2. As our results show, the UVB irradiation triggered skin pigmentation (B), and the hip skin around tail (circle in the photograph) was especially significantly pigmented. This circle area was subjected to each treatment. The photographs of the mice treated with NCP (D) showed slightly reduced pigmentation, and this de-pigmenting effect of NCP was quite similar to that of the Vt. C treatment (C). Interestingly, the mice treated with NCP in combination with Vit C (E) showed the fastest depigmentation effect.

### 3.2. Assessment of Melanin Index (MI) and Erythema Index (EI) Value

For measuring the anti-pigmentation effect of each treatment methods in a numerical manner, MI values were measured at just before the treatment (before) and immediately after the first (day 1), second (day 3), and third (day 6) of treatment. As the results of the MI value measurement shows (Figure 3A), UVB irradiation triggered the increase in MI value in all four UVB-irradiated mice. In contrast to the UVB + PBS-treated mice, which showed the slowest decrease in MI value (a +6-, −13-, and −28-point decrease on days 1, 3, and 6), the decrease in the MI value in NCP-treated mice was much faster (a −48.3-, −77.7-, and −88-point decrease on days 1, 3, and 6, respectively). Since the skin treated with Vt. C also showed a −40.3-, −67.3-, and −74.3-point decrease on days 1, 3, and 6, respectively, these data represent the fact that anti-melanin activity of NCP is similar to that of Vt. C. Interestingly, the mice that were subjected to the NCP + Vt. C combination treatment showed the fastest decrease in the MI index (a −67-, −103.7-, and −106-point decrease on days 1, 3, and 6).

Since UVB treatment also can trigger skin erythema, the EI value was also monitored (Figure 3B). As our results show, the EI value of the UVB-irradiated mice treated with PBS decreased by −3, −19, and −39 points on days 1, 3, and 6, respectively. On the other hand, the EI value of NCP-treated mice decreased by −18.3, −62.3, and −87.3 points on days 1, 3, and 6, respectively, showing that the anti-erythema effect of NCP was similar to that of Vt. C (a −18.3-, −41.3-, and −77-point decrease on days 1, 3, and 6). The significantly enhanced anti-erythema effect was seen in the mice treated with NCP + Vt. C, showing a decrease in EI value of around −45, −104.3, and −111 points on days 1, 3, and 6, respectively.

### 3.3. Histological Evaluation

Melanin pigment can be observed in the tissue through H&E staining. As our Figure 4 shows, the UVB-irradiated skin (B) displays several brown pigments within the epidermal tissue; however, only a few pigments were observed in the skin tissue of the UVB + Vt. C and UVB + NCP groups. Interestingly, in the UVB + NCP + Vt. C group, melanin pigments were not found. Furthermore, any significant skin damage caused by the Vt. C, NCP, and Vt. C + NCP treatments was not observed, but in the case of skin tissues treated with NCP, the epidermis was slightly thickened.

To visualize the effect of each treatment method on the recovery of UVB-damaged collagen, MST staining was performed (Figure 5). As our results show, UVB irradiation triggered the decrease in collagen density and the fragmentation of collagen (B). Interestingly, the tissues of all groups of mice treated with Vt. C (C), NCP (D), and NCP + Vt. C (E) showed collagen-recovering activity, and the collagen density of these three groups was even denser than that of the non-UVB-treated control.

IHC for tyrosinase, a key enzyme in the melanin synthesis process, was also performed (Figure 6) in order to visualize the effect of the NCP treatment on melanin synthesis in skin tissues. As our results show, the skin of mice treated with UVB + PBS (B) had increased tyrosinase protein within the epidermal tissue. In contrast to the skin of mice treated with UVB + Vt. C (C), which had similar tyrosinase protein levels to UVB + PBS, the UVB + NCP-treated skin tissue (D) showed a relatively decreased tyrosinase protein level. As the staining results of the UVB + NCP + Vt. C-treated mice (E) show, the combinational treatment of NCP with Vt. C completely recovered UVB-mediated increase in tyrosinase protein.

As a last effort for visualizing the anti-UVB effect of NCP, the changes in the melanin pigmentation of the skin tissues were monitored further via FMS staining. As Figure 7 shows, the skin irradiated with UVB has several black spots in the epidermis, and the skin treated with Vt. C shows a decreased number of black spots, with few of them still present. On the other hand, in the skin tissues treated with NCP and the tissue treated with NCP in combination with Vt. C, there were no melanin pigments detected.

## 4. Discussion

The focus of the skin care market concerns wrinkles, sunspots, whitening, lifting, and pyoderma. In particular, abnormalities such as sunspots or spots, pigmentation, and skin aging caused by the excessive synthesis of melanin due to active oxygen generated through exposure to UV rays are becoming a problem [21,22]. Since 2010, many products such as far infrared rays, ultrasound, and galvanic ions as well as supplemental devices for drug delivery, such as lasers, high frequency and LED devices were utilized in cosmetic medical devices for hospitals and clinics have been released [23]. Recently, non-invasive or minimally invasive devices are increasingly preferred in the market to minimize procedure costs, recovery periods, and side effects [24].

Ultraviolet rays have the greatest effect on the skin, and when skin is strongly ex-posed to UV, not only is excessive melanin formed but erythema, edema, sunburn, hyperplasia, inflammation, DNA damage, and photo-aging also occur. Severe exposure to UVB can lead to skin aging and skin cancer as well as hyperpigmentation. Recently, interest in substances that inhibit the production of melanin, in addition to UV blockers, is increasing in efforts to reduce skin damage caused by UVB exposure [25]. Vt. C plays an important role in whitening by maintaining the synthesis and elasticity of collagen in the dermis and preventing melanin from being pigmented on the skin. However, Vt. C can be easily oxidized and destroyed by heat, air, metal, and light [26]. The efficacy of skin whitening through Vt. C serum is proportional to its concentration [27]. Therefore, there is a requirement for additional tools to help the action of Vt. C.

In this study, NCP was adopted to test whether NCP could be a new tool for skin whitening by helping the action of Vt. C. For this, the melanin-containing HRM-2 hairless mice were subjected to UVB irradiation, and then the whitening effects of topical treatment of NCP, Vt. C, and NCP + Vt. C were tested. The photographs showing the effects of each treatment on external change of the skin (Figure 2) demonstrated that the treatment of NCP on UVB-irradiated mouse skin effectively improved pigmented skin. The results of MI measurement (Figure 3A) also support the results in Figure 2, demonstrating that treatment with NCP decreased the MI value significantly, and its effectiveness was similar to that of the Vt. C treatment alone. Although the MI value was decreased through Vt. C treatment, the change was not statistically significant. Interestingly, the MI value of mice treated with NCP + Vt. C showed a more dramatic decrease than Vt. C-treated mice. These data represent the fact that the topical application of NCP can be helpful for reducing UVB-induced skin pigmentation, and the combinational use of NCP with Vt. C can be more powerful than Vt. C treatment alone. According to Kim et al., the treatment of the NCP on melanocyte can inhibit the synthesis of melanin by decreasing tyrosinase activity [28]. In a line with this study, the fact that the topical application of NCP on UVB-irradiated mice also showed a skin whitening effect represents the possibility that the anti-melanocyte activity of NCP can be delivered through the epidermal tissues. The decreasing effect of the combinational treatment of NCP with Vt. C on the MI value might be the result of two factors. First, since NCP has melanin production-blocking activity, its effectiveness was added to that of Vt. C. The second possibility could be that the NCP’s transdermal drug delivery activity is helpful for the absorption of Vt. C, and thus elevated anti-melanin activity was observed.

The results showing the effect of NCP on the EI value (Figure 3B) also proved the fact that NCP can be helpful for UVB-mediated erythema. The mice treated with NCP displayed a significant decrease in EI value, and the mice treated with Vt. C also showed a significant decrease. Just like the result of MI value, the mice treated with NCP in combination with Vt. C showed the most dramatic decrease in EI value. In our previous report, the anti-inflammatory activity of NCP was elucidated using an atopic dermatitis mouse model [20]. This anti-inflammatory activity of NCP might be one of the reasons for the decreasing EI value in this study, since UVB irradiation also can cause dermal inflammation [29].

Meanwhile, tyrosinase is involved in the biosynthesis process of melanin, which causes skin pigmentation, with the first step in melanogenesis being induced by tyrosinase [30]. Melanin is synthesized through the enzymatic and non-enzymatic oxidation of tyrosine in cells called melanocytes located in the basal layer of the skin epidermis. The synthesized melanin in melanocytes can be transferred to the nearby keratinocytes, forming a melanosome unit [31]. In this study, the tyrosinase inhibitory activity of NCP was also tested by performing IHC. As Figure 6 shows, the skin of mice treated with NCP has reduced tyrosinase protein levels. Since the mice treated with Vt. C showed a mild decrease in tyrosinase protein, these data represent the fact that NCP might be more powerful for decreasing tyrosinase protein expression than Vt. C. Furthermore, since the NCP + Vt. C-treated mice showed the most decreased tyrosinase expression level, these data are also in line with the results shown in Figure 3 in this study.

The amount of melanin in the skin tissue is determined by the protein level and the activity of tyrosinase. The UV-mediated melanin synthesis can cause problematic pigmentation in the skin, accelerates skin aging, and can also cause skin cancer [32]. For this reason, melanin synthesis inhibitors have been studied, but problems regarding skin safety and formulation stability have been encountered [33]. Consequently, the need for research on safe and efficient melanin synthesis inhibitors has emerged [34]. In this study, the presence of melanin pigments in the skin tissue was visually monitored through FMS. As Figure 7 shows, the significant increase in melanin pigment was observed in UVB-irradiated tissue, but the skin pigmentation decreased upon NCP exposure, and the combinational use of NCP with Vt. C also reduced the melanin pigment in the skin tissue. These results demonstrate that the topical use of NCP on UVB-irradiated skin can be helpful for skin whitening.

The results of this study demonstrated that the topical application of NCP on UVB-irradiated skin inhibits tyrosinase level and consequently reduces the synthesized melanin pigment. Based on the results of this study, the significant skin whitening effect of NCP is expected; however, before the use of NCP is implemented in clinical and cosmetic practice, the verification of plasma stability and the ease of use is required in future research.

## 5. Conclusions

NCP does not cause histological damage to skin tissue, and the combinational treatment of NCP with Vt. C effectively reduced the pigmented skin of UVB-irradiated mice. This phenomenon might be caused by NCP’s inhibitory effect on the UVB-mediated accumulation of tyrosinase in the epidermal tissue, which is directly linked to the decrease in the melanin content of the skin. Taken together, this study demonstrates the outstanding characteristics of NCP as a new tool for treating pigmented skin.

## Figures and Tables

**Figure 1 biomedicines-11-01563-f001:**
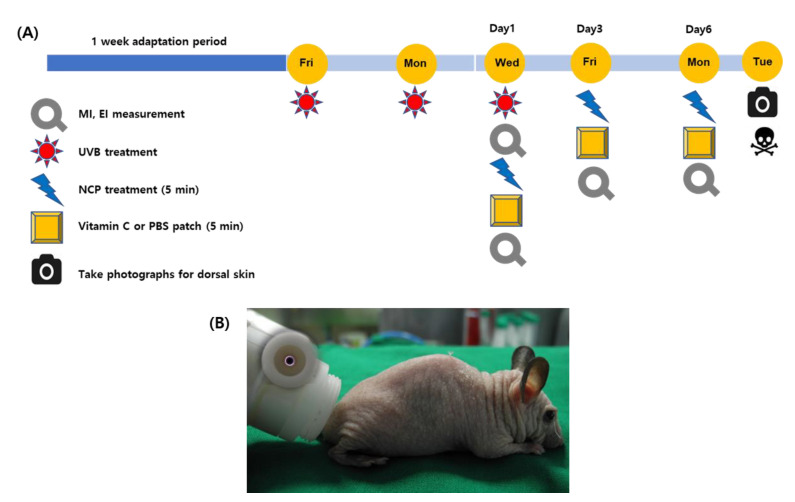
(**A**) The schematic diagram describing animal experiment procedures. (**B**) The NCP device forming a plasma plume within the device (small circle) and its application onto mouse skin.

**Figure 2 biomedicines-11-01563-f002:**
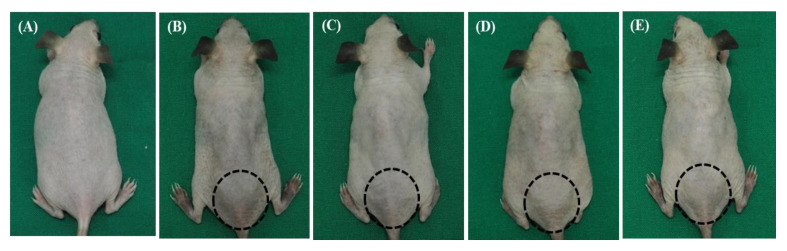
Photograph of the dorsal skin area showing depigmenting effects after 1 week of each treatment on UVB-induced pigmentation. (**A**) Control (no treatment); (**B**) 300 mJ/cm^2^ UVB + PBS; (**C**) 300 mJ/cm^2^ UVB + 10% Vt. C; (**D**) 300 mJ/cm^2^ UVB + NCP + PBS; (**E**) 300 mJ/cm^2^ UVB + NCP + 10% Vt. C.

**Figure 3 biomedicines-11-01563-f003:**
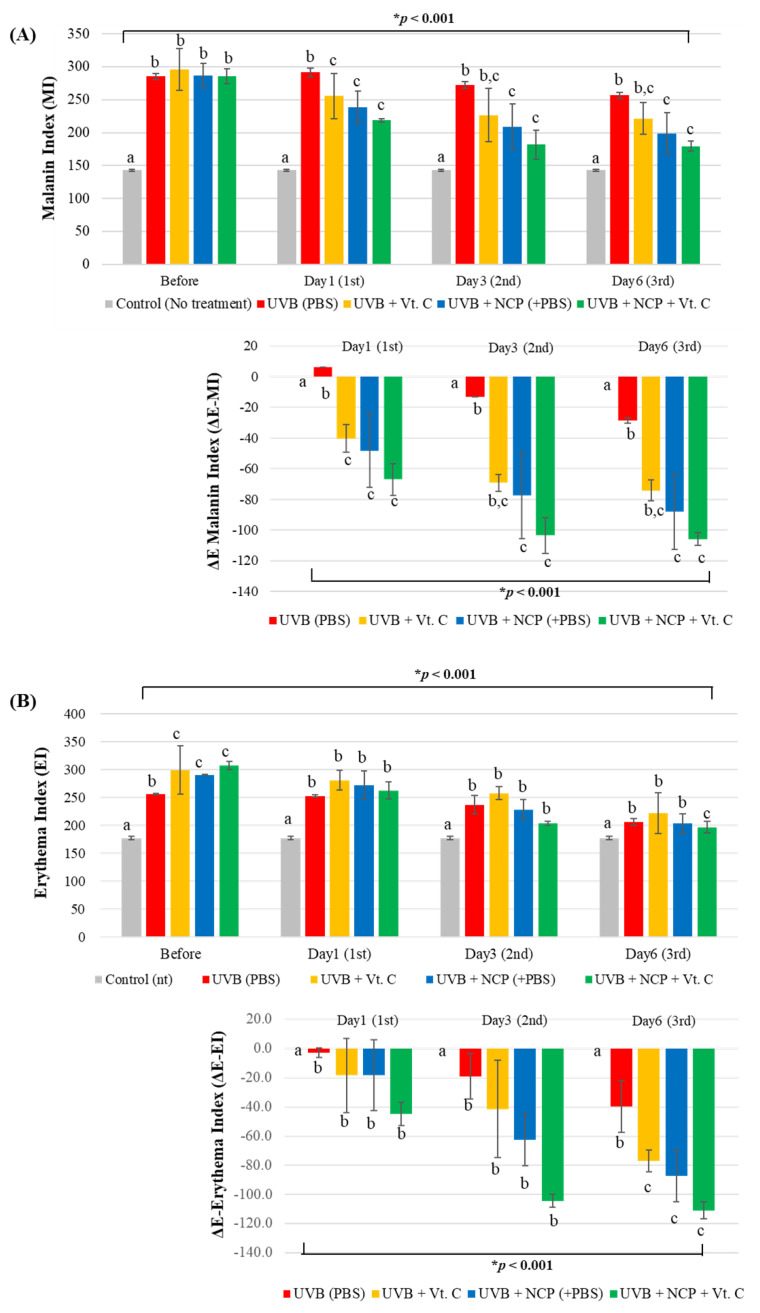
(**Top panel**) The graphs show the results of the melanin index (MI (**A**)) and the erythema index (EI (**B**)) value measurements using a Mexameter MX 18 probe from a MPA 5 device. (**Bottom panel**) The results show the changes in MI and EI values after each treatment (ΔE MI and EI). Different letters (a, b, and c) indicate the statistically significant results of the one-way ANOVA and Duncan’s post hoc test (* indicate statistical significance; *p* < 0.001).

**Figure 4 biomedicines-11-01563-f004:**
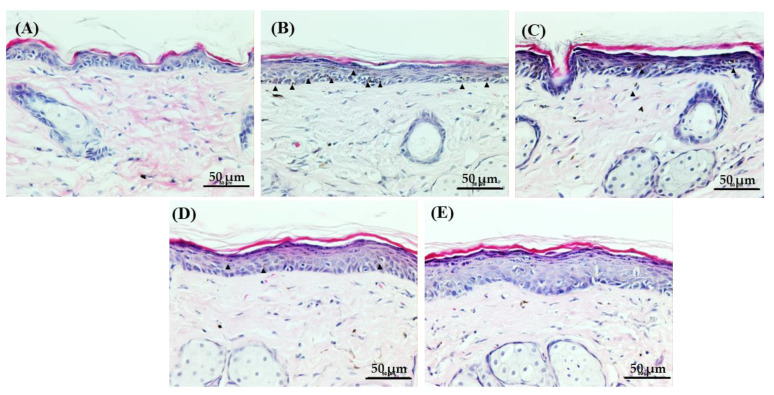
Histological characteristics of H&E staining. (**A**) Control (no treatment); (**B**) 300 mJ/cm^2^ UVB + PBS; (**C**) 300 mJ/cm^2^ UVB + 10% Vt. C; (**D**) 300 mJ/cm^2^ UVB + NCP + PBS; (**E**) 300 mJ/cm^2^ UVB + NCP + 10% Vt. C. The magnification of all images shown is ×200; scale bar: 50 μm. Small black triangles: brown pigments.

**Figure 5 biomedicines-11-01563-f005:**
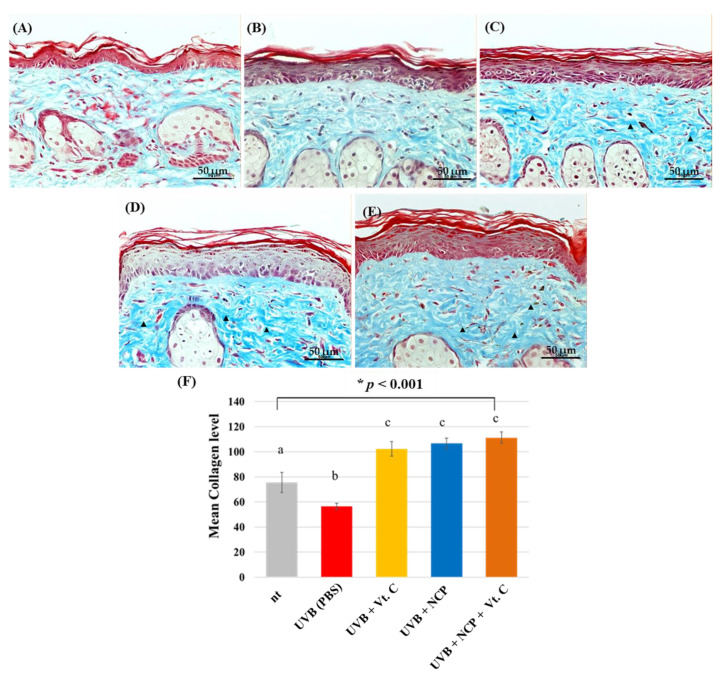
Histological response after MTS. (**A**) Control (no treatment); (**B**) 300 mJ/cm^2^ UVB + PBS; (**C**) 300 mJ/cm^2^ UVB + 10% Vt. C; (**D**) 300 mJ/cm^2^ UVB + NCP + PBS; (**E**) 300 mJ/cm^2^ UVB + NCP + 10% Vt. C. The magnification of all images shown is ×200; scale bar: 50 μm. (**F**) The graph shows the measurement of collagen density using Image J software. Different letters (a, b, and c) indicate the statistically significant results of the one-way ANOVA and Duncan’s post hoc test (* indicate statistical significance; *p* < 0.001).

**Figure 6 biomedicines-11-01563-f006:**
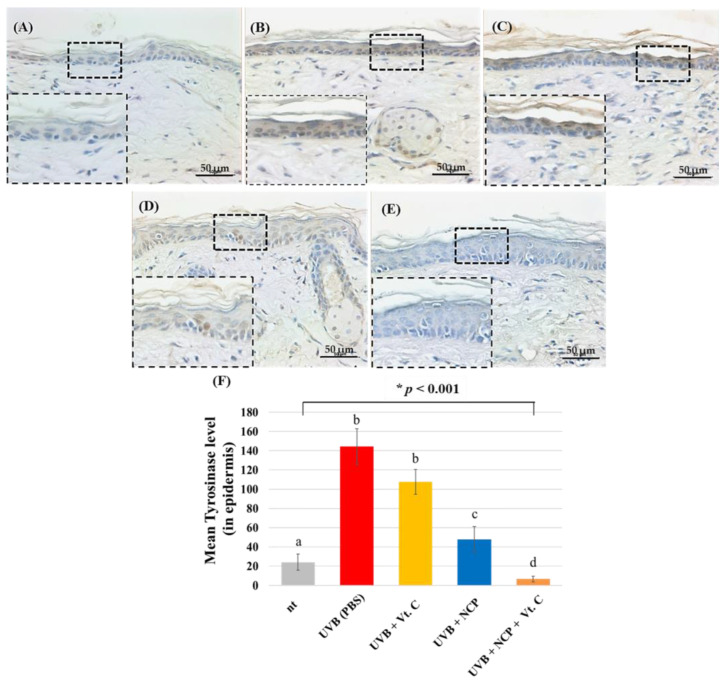
Histological characteristics and immunohistochemical detection of tyrosinase via IHC staining. (**A**) Control (no treatment); (**B**) 300 mJ/cm^2^ UVB + PBS; (**C**) 300 mJ/cm^2^ UVB + 10% Vt. C; (**D**) 300 mJ/cm^2^ UVB + NCP + PBS; (**E**) 300 mJ/cm^2^ UVB + NCP + 10% Vt. C. The magnification of all images shown is ×200; scale bar: 50 μm. (**F**) The graph shows the measured tyrosinase density in the epidermal tissue using Image J software (version 1.54c, NIH). Different letters (a, b, c, and d) indicate the statistically significant results of the one-way ANOVA and Duncan’s post hoc test (* indicate statistical significance; *p* < 0.001).

**Figure 7 biomedicines-11-01563-f007:**
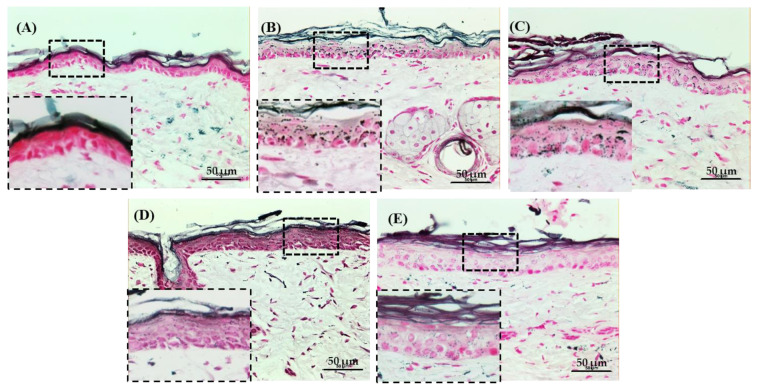
Detailed histological analysis of melanin content by FMS. (**A**) Control (no treatment); (**B**) 300 mJ/cm^2^ UVB + PBS; (**C**) 300 mJ/cm^2^ UVB + 10% Vt. C; (**D**) 300 mJ/cm^2^ UVB + NCP + PBS; (**E**) 300 mJ/cm^2^ UVB + NCP + 10% Vt. C. Magnification of all images shown is ×200; scale bar: 50 μm.

## Data Availability

Not applicable.

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
