# Peer review of "Topical Application of No-Ozone Cold Plasma in Combination with Vitamin C Reduced Skin Redness and Pigmentation of UV-Irradiated Mice"

_biomedicines, 2023, doi:10.3390/biomedicines11061563_

Round 1
Reviewer 1 Report (Previous Reviewer 3)
The authors took into account all my comments during the last review, except for one. The manuscript does not contain information about the installation generating plasma. There is no information about the type of plasma, its characteristics. Without this information, the results of the manuscript cannot be repeated. This needs to be fixed!
Author Response
Dear Reviewer 1,
Thanks for your kind comment.
Our point-by-point responses to your comment can be found in the attatched word file.
We hope that our resonses can be meet your expectation.
Sincerely yours,
Gyoo-Cheon Kim

Reviewer 2 Report (New Reviewer)
Starting by the end, this manuscript suggests that the topical use of o-ozone cold plasma in combination with vitamin C on UVB-irradiated skin can be helpful for skin whitening as reduces the melanin pigment in the irradiated region of nude mice. These results could be interesting, although before its application in clinical and cosmetic practice, verification of plasma stability and ease of use is required, as the authors rightly write at the final conclusion. I agree with these statements.
However, before acceptance, I recommend that the manuscript would be modified according to the following points:
a) Some methodological aspects should be improved:
1) It is stated that the MI and EI of the target dorsal skin were measured using Mexameter® MX18 (Courage + Khazaka electronic GmbH, Köln, Germany). The measurements were expressed as “points” but this is poorly informative. These units should be defined. At least, more details should be given for answering questions such as How is the scale determined? Are 100 points the full scale?.
At line 153: “Masson’s trichrome staining (MTS), or three-color staining in histology”.
Please, give more details about this protocol and how the three colors are generated. Otherwise, give a reference of the staining.
b) On the other hand, there are some “basic” statements about tyrosinase and melanogenesis pathway that are not really correct, so that I recommend they should be deleted or modified.
Line 41: Delete “promoting tyrosine synthesis by tyrosinase”. Tyrosinase is able to oxidize tyrosine to melanin, but it is not able to synthesize tyrosine.
Lines 53-54: Rewrite the sentence: “Vt. C not only removes free radicals by preventing oxygen from reacting with free radicals, but also restores free radical-mediated DNA damaged [9]”. It does not mind ref. 9, but vitamin C does not repair damaged DNA.
Line 59: Delete the sentence “anti-oxidative properties of Vt. C directly linked to the tyrosinase activity”, as the anti-oxidative properties of vitamin C are not linked to the tyrosinase activity, but the reduction of dopaquinone and indolequinone, ulterior steps in melanogenesis that are not directly linked to tyrosinase activity. Vitamin C can have other side effects, but this is too complex to be discussed in this particular work.
Line 62: Define SC (subcutaneous?)
Line 85: Define smps
Lin 91: Define kVpp
Line 97: “few OH radicals are generated from the device”. Indicate the source of the OH radicals and how these radicals were detected. According to line 59, vitamin C can generate OH radicals in specific conditions and presence of copper ions, favoring Fenton reaction. Generation of OH radicals could be very important in the particular context of this work.
Lines 291-292: delete the sentence “only products which contain more than 20% of Vt C was significantly effective for skin whitening [27]”. It does not mind ref. 27 (dated in 1999). As a matter of fact, at this manuscript, lines 121-122, it is stated that “for group 3, the mice skin was treated with a patch containing 10% vitamin C (Vt. C; L-Ascorbic acid 99%, Sigma-Aldrich, Sigma-Aldrich, St. Louis, MO, USA)”. Some references are useless, or worse than useless.
Line 313: Re-write the expression “NCP has anti-melanin activity”. Melanin has no activity.
Lines 357-358: Re-write the sentence “This phenomenon might be caused by NCP’s inhibitory effect on tyrosinase expression in the epidermal tissue”. This article suggest that the treatments show a hypopigmenting effect, either by inhibition of tyrosinase activity or melanin distribution through melanocytes and cutaneous keratinocytes, but there are no data to conclude anything about tyrosinase expression. Inhibition of tyrosinase expression involves blocking the expression of the tyrosinase gene.
c) Other points to be addressed.
Line 219: Please, explain why these results surprise.
Line 251: About the recovering of tyrosinase protein. How is that possible? Tyrosinase could be inactivated, but it is unlikely that tyrosinase as protein disappears. On the other hand, how does the indicated decrease reconcile with data at Figure 6?
Author Response
Dear Reviewer 2,
Thanks for your kind and valuable comments on our study.
Our point-by-point responses to your comments can be found in the attatched file.
We hope that our responses to your comments can be meet your expectations.
Sinserely yours,
Gyoo-Cheon Kim

This manuscript is a resubmission of an earlier submission. The following is a list of the peer review reports and author responses from that submission.
Round 1
Reviewer 1 Report
The authors reported that NCP alone possessed skin whitening effect through inhibiting tyrosinase activity without causing histological damage to skin tissue upon exposure. They also reported that NCP and Vt. C in combination resulted in greater skin whitening effects, probably through improved Vt. C penetration efficiency in the skin. The findings collectively suggest that NCP could be applied as an effective and safe treatment option for skin pigmentation. The overall study design was logical and experiments were performed in well-controlled manners. However, description of the experimental results needs improvement and clarity. Figure 3 is confusing. One would expect that that both MI and EI values should be increased in the UVB+PBS samples compared to the non-UV treatment control. The statistical labels are also difficult to follow. Also, tyrosinase expression in Figure 5E (combo treatment) is comparable to the UVB+PBS alone (5B) and higher than NCP or Vt. C alone (5C & 5D). How would the authors explain the strongest inhibition of UVB-induced pigmentation by combo treatment despite that there is less inhibition of tyrosinase expression? The studies focused on UVB irradiation, but the title referred to “…phenotypes of UV-irradiated mice skin”. Is NCP effective in inhibiting UVA-induced skin pigmentations as well? The paper would benefit greatly from extensive language editing, which will help readers to follow the studies more easily.
Author Response
"Please see the attachment."

Reviewer 2 Report
Paper titled "No-ozone cold plasma treatment in combination with vitamin C improved the phenotypes of UV-irradiated mice skin" by Nam et al. studied the benefit of cold plasma and vitamin C in improving the UV irradiated mouse skin. They claimed that this combination is effective in protecting against UV irradiation on mouse skin. The paper has many points of weakness that must be corrected before further consideration.
1- Title: needs to be revised to be more informative & tell how was this improvement. Also mention the route of application or administration. (oral, topical,...etc)
2- Abstract should be amended by some numerical values
3- Try to shorten the Introduction to be more concrete
4- Aim of the study at the end of INTRO need to be separated to be explored. Confirm about the aim & how it was achieved.
5- What was the age & weight of the animals at the begin of the experiment
6- USe appropriate abbreviations for minutes, seconds, hours..etc.
7- A graph for the design of the experiment will be a valuable addition
8- Authors should give the source of chemicals, kits and antibodies completely and consistently (code, company, town, state and country) & version for software
9- 2.4. Histological and immunohistochemical analysis 174 The skin samples from each group: from which part of the animal?
10- Describe in methods ho w" melanin index (MI) and Erythema index (EI) value" were calaculated precisely
11- Authors claimed "The melanin and erythema indices were analyzed by one-way analysis of variance 200 (ANOVA) and post-hoc Tukey’s test"" :9- Authors have to check the normality of distribution of the results by a suitable post hoc test (such as Shapiro-Wilk test or K-S test) before deciding to choose certain ANOVA. If the normality test indicated normal dist of the data, so use one-way ANOVA, if not, use non parametric ANOVA.
12- Figure 3 is comletely diffciult to be comprehended, this is not a common style for presentating stat sifference. We do not repeate the significance sympols & each difeerence is usually indicated for once.
13- At the figure legend, significance sympols (a,b,c,d,) should be explained
14- Figure 4 &5 : should contain arrows to refer to pathological findings
15- immunohistochemistry in figure 6 should be quantified by image analysis software & values should be compared statistically. ALso figur e 5
16- Conclusion is Worng": improved penetration efficiency???!!! no evidence on this part
17- Unless stat analysis is improved in figures & authors should confirm in methods "every possible comparison was done " & apply this to column charts.
18- How the plasma was prepared or applied volume should be indicated
19- Quality of images IHC in figure 6 is very poor & should be replaced with high resolution images
20- It would be better to mention power & scale bar for each image
21- Method sin general lacks refernces
Author Response
"Please see the attachment."

Reviewer 3 Report
Manuscript review "No-ozone cold plasma treatment in combination with vitamin C improved the phenotypes of UV-irradiated mice skin" by Seoul-Hee Nam, Jeong-Hae Choi and Gyoo-cheon Kim. An interesting and relevant manuscript, but in the process of reading I found a number of problems.
1. For me, the facts that the authors cite when motivating the study remained unclear. For example, the authors write: "Today, as the demand for white and transparent skin increases and the need for skin whitening products increases, research on skin whitening is continuously being conducted". I see hundreds and thousands of people sunbathing in the sun on the beaches, on the lawns in the parks, probably the opposite, the society has a demand for tanned skin. At least in European society. Perhaps the authors need to slightly modify their statement and clarify in which society there is a request about which they write.
Link 9 is not true and cannot be found on the Internet! In general, vitamin C is not involved in DNA repair, however, in the presence of variable valence ions, it can lead to damage to DNA bases and detachment of sugar from the nucleotide.
The authors write that vitamin C is included in a large number of creams, while it should be noted that vitamin C creams are usually very low!
The authors write that CO2 lasers are not very good, and not laser sources in the visible range are good. This is not a correct comparison and what an absurdity! Carbon dioxide lasers emit in the infrared range, with a wavelength of 9.4 to 10.6 microns. Melanin does not absorb in this range, but absorbs in the UV and visible range...
The authors have a rather unconventional opinion about plasma physics ...
Plasma damages the skin quite effectively, so prolonged exposure to cold plasma can be considered damaging.
After reading the introduction, it is not at all clear why the authors decided to use cold plasma. This question needs to be answered!
2. The device for obtaining plasma is not described, the physico-chemical characteristics of the plasma are not presented. The experiments presented in the manuscript cannot be repeated.
3. The results presented are not entirely clear and in this form are ambiguous. So in Figure 3 it is not at all clear what the three repeating groups of histograms mean. Different time after processing? Different number of treatments? Some other variations?
About Figure 4, the authors write that there are pigment spots only in Figure B, while there are spots on C, D and E. Some kind of numerical representation is needed. These photos do not convince!
The following figure, the authors say all methods restore the amount of collagen. Indeed, in the control and after irradiation there is little collagen, in cases of treatment there is a lot. What does it mean? Normal animals have little, after irradiation little, and then a lot. Maybe it's not good?
As for the Histological characteristics and immunohistochemical detection for Tyrosinase, I did not understand at all on the basis of what the authors draw their conclusions.
The last picture is also not convincing!
So, in fact, we have an ambiguous introduction, a section of materials and methods in which there is no key information and ambiguous results consisting of an automatic measurement of two indices and four types of histology. The verdict, the manuscript needs to be significantly improved before publication!
Author Response
"Please see the attachment."

Round 2
Reviewer 2 Report
The revised form of paper {No-ozone cold plasma treatment in combination with vitamin C improved the phenotypes of UV-irradiated mice skin} was moderatly improved & some points are still need correction:
1-Comment 16: 1. authors should confirm in methods "every possible comparison was done " & apply this to column charts: this means you should compare between each 2 groups and give sympols if p<0.05 on the column chart.
2- Authors aid that they checked normality in the reply letter, but the name of the test & this confiration did not appear in the methods section
3- Figure 3 (all panels) & other column charts should be improved by increasing thickness of x and y axis & enlarge the chart & improve resolution, the current format is unacceptabe.
Author Response
"Please see the attachment."

Reviewer 3 Report
The manuscript has been greatly improved. I have doubts about a number of formulations, but these doubts are not significant. Before publication, I suggest that the authors add in the form of a small paragraph or table the main physical properties of plasma and plasma-treated objects. You do not need to send me the manuscript for further consideration.
Author Response
"Please see the attachment."

Round 3
Reviewer 2 Report
Unfourtnately, the revised form of this paper did not reply adequately to the comments:
1- The authors wrote that " The melanin and erythema indices, the results of quantification of collagen and ty- 174 rosinase level from the histological images were analyzed by one-way analysis of variance 175 (ANOVA) and post-hoc Tukey’s test. For all case, the normality of the distributions was 176 assessed with the Shapiro-Wilks test and p value <0.05 was considered statistically signif- 177 icant."
This indicates they do not understand what normality test means !! as it should be applied before deciding to using what type of ANOVA!! It seems they just write the sentence to satisfy the reviewer but did not apply the test actually.
2- Authors did not do every possible comparison in stat analysis although we asked them to do it twice. They just compared to 2 groups only. They have 5 groups.
3- The axis of the figures were not enhanced or increased in thickness to be clear as asked for this point twice before.'
Upon this NOT correct stat analysis, This reviewer cannot trust these data or conclusion drawn from this article. & it must be rejected.